# A Novel Inducible Prophage from *Burkholderia vietnamiensis* G4 Is Widely Distributed across the Species and Has Lytic Activity against Pathogenic *Burkholderia*

**DOI:** 10.3390/v12060601

**Published:** 2020-05-31

**Authors:** Rebecca Weiser, Zhong Ling Yap, Ashley Otter, Brian V. Jones, Jonathan Salvage, Julian Parkhill, Eshwar Mahenthiralingam

**Affiliations:** 1Cardiff School of Biosciences, Cardiff University, Cardiff CF10 3NB, UK; yapzl@myumanitoba.ca (Z.L.Y.); ashley.otter@phe.gov.uk (A.O.); 2Department of Microbiology, University of Manitoba, Winnipeg, MB R3E 0J9, Canada; 3Public Health England, National Infection Service, Porton Down, Salisbury SP4 0JG, UK; 4Department of Biology and Biochemistry, University of Bath, Bath BA2 7AY, UK; bvj20@bath.ac.uk; 5School of Pharmacy and Biomolecular Sciences, University of Brighton, Brighton BN2 4GJ, UK; j.p.salvage@brighton.ac.uk; 6Department of Veterinary Medicine, University of Cambridge, Cambridge CB2 0XY, UK; jp369@cam.ac.uk

**Keywords:** *Burkholderia vietnamiensis*, prophages, phylogenomics, induction, lysogeny, phage classification

## Abstract

*Burkholderia* species have environmental, industrial and medical significance, and are important opportunistic pathogens in individuals with cystic fibrosis (CF). Using a combination of existing and newly determined genome sequences, this study investigated prophage carriage across the species *B. vietnamiensis*, and also isolated spontaneously inducible prophages from a reference strain, G4. Eighty-one *B. vietnamiensis* genomes were bioinformatically screened for prophages using PHASTER (Phage Search Tool Enhanced Release) and prophage regions were found to comprise up to 3.4% of total genetic material. Overall, 115 intact prophages were identified and there was evidence of polylysogeny in 32 strains. A novel, inducible Mu-like phage (vB_BvM-G4P1) was isolated from *B. vietnamiensis* G4 that had lytic activity against strains of five *Burkholderia* species prevalent in CF infections, including the Boston epidemic *B. dolosa* strain SLC6. The cognate prophage to vB_BvM-G4P1 was identified in the lysogen genome and was almost identical (>93.5% tblastx identity) to prophages found in 13 other *B. vietnamiensis* strains (17% of the strain collection). Phylogenomic analysis determined that the G4P1-like prophages were widely distributed across the population structure of *B. vietnamiensis*. This study highlights how genomic characterization of *Burkholderia* prophages can lead to the discovery of novel bacteriophages with potential therapeutic or biotechnological applications.

## 1. Introduction

Bacteriophages are estimated to be the most abundant entities in the biosphere, and as the natural predators of bacteria, have important roles in bacterial ecology and evolution [1]. Lytic phages are capable only of infections ending in bacterial cell death and release of new phage progeny (lytic life cycle), whilst temperate phages can either behave as lytic phages, or integrate into the genome of a bacterial host as a prophage (lysogenic life cycle) [1]. Prophage carriage can confer a number of benefits to the lysogen including acquisition of novel traits through horizontal gene transfer, increased genetic variation and evolutionary innovation [2]. Resident prophages also present a risk for bacteria, as induction of the lytic cycle can occur spontaneously or in response to environmental cues (e.g., stress) and leads to cell lysis [2]. Prophages are abundant in many bacterial species and can represent up to 10–20% of the host’s genome [3]. Prophage carriage is very variable between bacteria, however, some bacteria have none, whilst others are polylysogenic and can carry over a dozen prophages [4]. Large-scale bacterial genome sequencing projects in parallel with the development of prophage finding software [5,6,7] have enhanced our ability to detect prophages and understand their distribution in a range of bacteria.

*Burkholderia* are a very diverse group of Gram-negative bacteria found in wide-ranging environments including soil, water and in association with plants, fungi, animals and humans [8]. The genus has undergone extensive taxonomic revision since its definition in 1992 [9], and has recently been split into six genera, *Burkholderia sensu stricto*, *Paraburkholderia*, *Caballeronia*, *Trinickia*, *Mycetohabitans* and *Robbsia* [10]. There are currently more than 30 *Burkholderia* species, 22 of which belong to the *Burkholderia cepacia* complex (Bcc) [10]. Members of the Bcc are opportunistic pathogens linked with life-threatening lung infections in individuals with cystic fibrosis (CF) [11] and are problematic contaminants of industrial products [12]. Conversely, Bcc bacteria have numerous beneficial traits including antimicrobial production, plant growth promotion and bioremediation [8]. The large (6–9 Mb), multi-replicon genomes of *Burkholderia* have a huge coding capacity that underpins their genetic and metabolic versatility, and an ability to thrive in different environments [13]. Whilst Bcc species are closely related by taxonomic analysis based on a single marker gene (>97.7% 16S rDNA sequence similarity) [14], recent comparative genomic studies have identified huge genomic diversity within the complex; there are approximately only 1000 genes shared between strains of the 22 Bcc species (the “core genome”), highlighting that a large proportion of the genome is highly variable [15]. *Burkholderia* genomes have also been shown to comprise up to 10% recently horizontally acquired DNA [16] and potentially represent a rich source of prophages. *Burkholderia* phage literature is relatively limited, however, and although phage therapy for the treatment of Bcc infections is being explored [17], there have been few systematic studies looking at prophage carriage within *Burkholderia* genomes.

*B. vietnamiensis* is part of the Bcc and known to cause infections in individuals in CF. It is encountered less frequently than the two major Bcc CF pathogens, *B. cenocepacia* and *B. multivorans* [18], and the prevalence of *B. vietnamiensis* within overall *Burkholderia* CF lung infections is generally under 10% [11,18,19,20,21,22]. *B. vietnamiensis* has been associated with non-CF clinical infections [18,21], but also has strong links to the environment, being isolated from soil and plant roots, having nitrogen fixation abilities [23,24] and the capacity to degrade a range of xenobiotic compounds [25]. There are currently 48 *B. vietnamiensis* genomes available in Genbank (accessed on 31 January 2020; https://www.ncbi.nlm.nih.gov/genome/genomes/1136?#) averaging 6.86 Mb in size (range: 5.73–8.39 Mb). The largest genome belongs to *B. vietnamiensis* strain G4 (ATCC 53716) whose 8.39 Mb genome is organised into 3 chromosomes and 5 plasmids. This strain was originally isolated from wastewater in 1987 and is well known for its ability to degrade trichloroethane when grown on toluene or phenol [26,27]. *B. vietnamiensis* genomes have not yet been investigated for prophage carriage and only one lysogenic phage has been previously isolated from *B. vietnamiensis* ATCC 29424 [28].

We have taken advantage of the increasing numbers of available *B. vietnamiensis* genomes, supplemented these with 35 sequenced for this study, and used them to characterise prophage regions within the species. We report that prophage material ranged from 0.1% to 3.4% of *B. vietnamiensis* genomes and polylysogeny was observed. In addition, a novel inducible and functional prophage (G4P1) was discovered in *B. vietnamiensis* G4 that had lytic activity against strains of five Bcc species. Phage G4P1 was widely distributed across the population structure of the species and found in 17% of the 81 *B. vietnamiensis* strains examined. Our investigations linked genomic and phylogenomic approaches to characterize prophage carriage across the species, and facilitate the discovery of novel phages with potential therapeutic or biotechnological applications.

## 2. Materials and Methods

### 2.1. Bacterial Genomic Analysis and Phylogenomics

A total of 35 *B. vietnamiensis* strains were genome sequenced using the Illumina HiSeq 2000 and HiSeq X Ten platforms to generate 125-nucleotide and 150-nucleotide paired end reads, respectively, as described by Mullins et al. (2019) [29]. The raw read data have been submitted to the European Nucleotide Archive (ENA; sequencing project PRJEB9765). Individual accession numbers are given in Appendix A. Additional *B. vietnamiensis* genomes were downloaded directly as draft assemblies (FASTA sequences) from Genbank (*n* = 46) (Appendix A). There were two entries for strain FL_5_2_10_S1_D0 in Genbank which were designated FL_5_2_10_S1_D0 (GCA_001524025.1) and FL_5_2_10_S1_D0_Repeat (GCA_001524045.1) throughout.

All bioinformatic analyses were performed using the Cloud Infrastructure for Microbial Bioinformatics (CLIMB) computing resource [30]. Raw reads were trimmed using Trim Galore v0.4.3 for paired end reads [31] and quality assessed with FastQC v0.11.5 [32]. Genome assembly was achieved using Unicycler v0.4.7 [33] with SPAdes v3.11.0 [34] and the option for short-read assembly. Assembly quality was visualised with Bandage v0.8.1 assembly graphs [35]. All genome assemblies were quality checked using the Quality Assessment Tool for Genome Assemblies (QUAST) v.4.6.3 to confirm expected genome size and key assembly statistics (Appendix A). *B. vietnamiensis* species identity was confirmed by calculating the shared average nucleotide identity (ANI) between genomes using FastANI v1.1 [36] to ensure that the 95% species threshold was met. Prokka v1.12 [37] was used for gene annotation. Roary v3.6.0 [38] and double-precision FastTree v2.1.8 [39] were used for comparative core genome analysis of *B. vietnamiensis* strains and generation of phylogenomic trees as described previously [40]. The phylogenomic tree root position was determined using the outgroup *B. ambifaria* AMMD (GCF_000203915.1; Appendix A). Manual inspection of *Burkholderia* genomes was performed using Artemis v17.0.1 [41] with the FNA and GFF file outputs from Prokka.

### 2.2. Prophage Identification and Comparison

Prophage regions were identified in *B. vietnamiensis* genomes using the PHASTER (Phage Search Tool Enhanced Release) online tool [5] [performed January 2019] which scores putative phage regions as either intact, questionable or incomplete. R statistical software [42] was used to visualise trends and perform linear regression analysis (lm function) to identify significant correlations (*p* ≤ 0.05 level) between the amount of prophage material carried and bacterial genome size. Comparative ANI analysis of prophages sequences identified as “intact” was performed using pyani v0.2.7 [43] with the ANIb option. The command-line version of NCBI BLAST v2.7.1 [44] was used to create and search (blastn, tblastx) local BLAST databases of intact prophage sequences with a prophage sequence query. Prophage genome alignments were performed using Easyfig v2.2.3 [45]. The GBK files for the visualisation of genome comparisons in Easyfig were obtained by annotation of intact prophage sequences using Prokka with the Millardlab *Caudovirales* custom annotation database (http://millardlab.org/bioinformatics/lab_server/phage-genome-annotation/).

Transposase proteins from *B. vietnamiensis* prophages (this study), related sequences identified by blastp (https://blast.ncbi.nlm.nih.gov/Blast.cgi) (results restricted within Myoviridae, taxid 10662), and other Mu-like *Burkholderia* phages (BcepMu, NC_005882.1; ΦE255, NC_009237.1), were compared by alignment using ClustalW [46] and Maximum Likelihood phylogenetic trees generated using MEGA7 (Jones–Taylor–Thornton model, with Neighbor–Join and BioNJ methods used to create initial trees for the heuristic search) [47,48].

### 2.3. Bacterial Strains and Culture Media

*Burkholderia* strains were obtained from the BCC culture collection held at Cardiff University [49] (Table 1). Strains were routinely grown on Tryptone Soya Agar (TSA; Oxoid Ltd., Cambridge, UK) and overnight cultures prepared by inoculating 3 mL of Tryptone Soya Broth (TSB; Oxoid Ltd.) with fresh (<72 h) growth material and incubated on an orbital shaker (150 rpm). All cultures were incubated at 30 °C for 16–18 h. Bacterial strains were stored frozen in TSB containing 8% (*v*/*v*) dimethylsulphoxide (DMSO; Sigma-Aldrich Company Ltd., Dorset, UK) at −80 °C.

### 2.4. Isolation of Spontaneously Induced Bacteriophages from Lysogenic Burkholderia vietnamiensis G4

An overnight culture of *B. vietnamiensis* G4 prepared in TSB + 10 mM MgCl_2_ was centrifuged at 2504 *g* for 10 min and the supernatant filter sterilised (pore size 0.2 µm; Sartorius Stedim Biotech, Cambridge, UK). Lytic phage activity was screened for against a panel of 24 strains representing 9 different Bcc species, 2 strains of non-Bcc *Burkholderia* species and 3 *Paraburkholderia* strains (Table 1) using the double agar overlay technique [50] and a “drop test”; a soft agar top layer (TSB + 0.3% (*w*/*v*) purified agar (Oxoid Ltd.) + 10 mM MgCl_2_) seeded with approximately 10^6^ viable host cells was overlaid on TSA and 10 μL culture supernatant dropped onto the surface. After overnight incubation at 30 °C for 16–18 h, lytic phage activity was identified as a zone of clearing in the overlay layer. Zones were classified as strong activity (complete clearing), weak activity (partial clearing) or no activity. Positive supernatants were re-tested.

Plaque assays [51] with appropriate host strains were used to identify individual lytic bacteriophages in culture supernatants and produce phage stocks. The host strains used were *B. ambifaria* BCC1212 (phage G4P1) and *B. cenocepacia* BCC1210 (phages G4P2 and G4P3). Low titer individual phage stocks were prepared by transferring a single plaque into 5 mL phage buffer (Phosphate Buffered Saline (PBS; Oxoid Ltd.) + 10 mM MgCl_2_). The phage suspension was then vortex mixed for 1 min, centrifuged at 2504*g* for 10 min and filter sterilised (pore size 0.2 µm). To obtain high titer phage stocks (approximately 10^7^–10^9^ pfu/mL) plaque assays were performed with low titer phage stocks and agar overlays displaying semi-confluent lysis (≥4 agar plates) were transferred into 10 mL phage buffer, vortex mixed for 1 min, centrifuged at 2054× *g* for 10 min then filter sterilised (pore size 0.2 µm). Storage of culture supernatants and phage stocks was at 4 °C (short term) or at −80 °C with 50% glycerol (Fisher Scientific UK Ltd., Loughborough, UK) (long term).

### 2.5. Bacteriophage Host Range Determination of Phage G4P1 Isolated from B. vietnamiensis G4

A double layer agar overlay of each host strain (Table 1) was prepared and 10 µL of high titer G4P1 phage stock dropped onto the overlay surface. Following overnight incubation at 30 °C for 16–18 h, lytic phage activity was observed as a zone of clearing in the top agar layer. Positive results were confirmed by repeating the assay. Plaque assays were also performed with the G4P1 high titer stock and four of the host strains (*B. ambifaria* BCC1212, *B. cenocepacia* BCC0019, *B. dolosa* BCC1359 and *B. vietnamiensis* BCC0027) to identify individual phage plaques and confirm that the zone of clearing in the drop test was of phage origin rather than another inhibitory factor.

### 2.6. Transmission Electron Microscopy (TEM)

The morphology of phage virions was investigated by TEM as previously described [52]. Phage particles from high titer stocks were deposited onto 200-mesh Formvar/carbon copper grids (Agar Scientific, Essex, UK), and negatively stained with 2% phosphotungstic acid (pH 7.4) (Sigma-Aldrich Company Ltd.). Virions were imaged using a Zeiss Sigma field-emission gun-scanning transmission electron microscope (FEG-STEM; Zeiss, Cambridge, UK) at 20 kV accelerating voltage, a 20 µm aperture and a 2.7 mm working distance.

### 2.7. Bacteriophage Genomic Analysis

Bacteriophage DNA was extracted from high titer phage stocks using the Phage DNA isolation Kit (Norgen Biotek Corp., Ontario, Canada) according to the manufacturer’s instructions and with the recommended DNase I treatment step. A 16S rRNA gene polymerase chain reaction (PCR) [53] was performed to confirm that phage DNA stocks were free from contamination with host bacterial DNA. DNA quantification was achieved using a Qubit fluorometer and the Qubit dsDNA BR assay kit (Invitrogen, Massachusetts, USA).

DNA sequencing library preparation and sequencing was performed by the Cardiff University Genomics Research Hub. Libraries were prepared with the Nextera XT kit (Illumina, California, USA) and sequencing was carried out on an Illumina NextSeq500 using a NextSeq 500/550 Mid Output v2 kit (2 × 150 cycles) to give on average 122 bp paired-end reads. Read trimming, genome assembly and quality control was performed as described for bacterial genomes to produce a single contig (36725 bp). Coverage was determined by mapping the sequence reads onto the assembly using Samtools v1.7 [54] and Bedtools v2.25.0 [55] and was on average 442X (Appendix A; Appendix A). Gene annotation was achieved using a combination of Prokka with the Millardlab *Caudovirales* custom annotation database (as described for prophage annotation), PHASTER and the InterProScan tool of the online InterPro resource for further functional analysis of predicted protein sequences (https://www.ebi.ac.uk/interpro/search/sequence/) [56,57]. Isolated and genome sequenced bacteriophages were named according to recent recommendations [58].

To determine the genomic location of isolated phages as prophages within the lysogen genome, the command-line version of NCBI BLAST v2.7.1 [44] was used to create and search (blastn) a local BLAST database of the lysogen genome with a phage DNA sequence query. Easyfig v2.2.3 was used to visualise comparisons between phage and lysogen sequences, and the *Burkholderia* genome database [59] used for lysogen gene annotations.

### 2.8. Data Summary

All de novo determined genome sequence data for *B. vietnamiensis* is available at the European Nucleotide Archive under sequencing project PRJEB9765.

The authors confirm all supporting data, code and protocols have been provided within the article or through Appendix A.

## 3. Results

### 3.1. Prophage Carriage Was Common in B. vietnamiensis

To gain an understanding of prophage carriage in *B. vietnamiensis*, PHASTER was used to examine the genomes of the strain collection. All 81 *B. vietnamiensis* strains had evidence of prophage material and 59 of these had at least one intact prophage (Appendix A). Overall, 115 intact, 23 questionable and 81 incomplete prophages were discovered and the majority of strains carried one or two prophage regions of each type (Figure 1A). Intact prophage regions (average = 35.2 kb, range = 12.3–92.0 kb) were larger than questionable prophage regions (average =27.4 kb, range = 4.9–55.4 kb), which in turn were larger than incomplete regions (average = 17.8 kb, range = 5.6–48.6 kb) (Figure 1B). As genome size increased, the total prophage material carried also increased (Figure 1C; *p* < 0.001), and on average strain genomes comprised 1.1% (range: 0.1–3.4%) prophage material (Appendix A). Polylysogeny with intact prophage regions was observed in 32 strains; 18 strains had two intact prophages, 10 strains had three, 3 strains had four, one strain (BCC1193) had five and one strain (BCC1172) had six (Appendix A).

### 3.2. Prophages Were Spontaneously Induced from B. vietnamiensis Strain G4

*B. vietnamiensis* G4 was historically isolated as a bioremediation strain and examined as a model strain in biotechnological studies [25]. Although it was one of the first members of the Bcc to be fully genome sequenced, the prophage content of this strain has not been characterised. PHASTER predicted that the 8.39 Mb *B. vietnamiensis* G4 genome held two intact, two questionable and six incomplete prophage regions (Appendix A) which totaled 276 kb, representing 3.29% of the G4 genome. To determine if any of these regions were spontaneously inducible and able to form plaques, the supernatant of an overnight culture of *B. vietnamiensis* G4 was tested against a panel of 26 *Burkholderia* and 3 *Paraburkholderia* strains (Table 1). Lytic activity was observed against strains of 5 different Bcc species (*B. ambifaria*, *B. cenocepacia*, *B. contaminans*, *B. dolosa* and *B. vietnamiensis*) (Table 1) in a drop test assay.

To isolate individual phages, plaque assays were performed with the *B. vietnamiensis* G4 culture supernatant and the host strain *B. ambifaria* BCC1212. This resulted in clear plaques 0.5–1 mm in diameter (Figure 2a,b) and an individual plaque was isolated, the phage purified and designated G4P1. The host range of G4P1 was identical to that of the *B. vietnamiensis* G4 supernatant with the exception that it was not active against *B. cenocepacia* BCC1210 and had “extra” activity against *B. vietnamiensis* BCC1304. This suggested the presence of more than one inducible prophage with a different host range to G4P1, and that individual phages may behave differently when purified from culture supernatants. Additional plaque assays with the *B. vietnamiensis* G4 supernatant and *B. cenocepacia* BCC1210 as the host isolated at least one other phage; two different plaque morphologies were consistently observed and designated G4P2 (clear plaques 0.5–1 mm in diameter; Figure 2c,d) and G4P3 (uniform but slightly cloudy plaques 0.5–1 mm in diameter; Figure 2e,f). Only G4P1 stored stably at 4 °C without a loss in titer and was further characterised. TEM revealed that G4P1 was a tailed phage with an icosahedral head belonging to the order *Caudovirales* and was approximately 200 nm in length (Figure 2g). Extended host range testing with G4P1 determined that the phage had activity against 5 other strains of *B. dolosa* including the Boston epidemic strain SLC6 (Table 1).

### 3.3. G4P1 was Localised to Chromosome 1 of B. vietnamiensis G4

The DNA sequence obtained for G4P1 was 36725 bp in length and was compared to the *B. vietnamiensis* G4 genome to determine its location as a prophage. G4P1 shared 100% blastn identity with a 36,695 bp region on chromosome 1 starting in the intergenic region between tRNA-Arg (R0040) and spanning Bcep1808_1284 to Bcep1808_1331 (Figure 3). G4P1 encoded 49 genes, mobility by transposition and had clusters of early (non-structural; *n* = 17) and late (structural; *n* = 32) genes characteristic of Mu-like phages, named on the basis of their different levels of transcription during the lytic lifecycle (Figure 3; Appendix A). G4P1 was similar in organisation to the previously discovered *Mannheimia* Mu-like phages vB_MhM_3927AP2 and phiMhaMu2, and *Haemophilus* phage SuMu, having a late phage lysin gene located immediately downstream of the early transcription activator gene *mor*, and lacking middle genes such as *C* which are late gene transcription activators [60]. Approximately 45% of the G4P1 genes (*n* = 21) were designated as hypothetical proteins due to annotation as such or lack of concordance between annotation methods (Appendix A). These features, in addition to the morphological characteristics, placed the phage within the family *Myoviridae* and hence G4P1 was given the name vB_BvM-G4P1 (Virus of Bacteria, *B. vietnamiensis* lysogen, *Myoviridae*, phage G4P1) to account for these characteristics. PHASTER identified an intact prophage in this location as prophage region 10 (Figure 3; Appendix A), but inclusion of the Bcep1808_1284 gene (beta-galactosidase) was not predicted by the software, perhaps because this gene was likely to be of bacterial origin.

### 3.4. vB_BvM-G4P1 Was Found in Other B. vietnamiensis Strains

G4 PHASTER prophage region 10 (G4P1 without the beta-galactosidase gene) was compared to the 114 other intact *B. vietnamiensis* prophage regions identified by PHASTER. Highly similar regions were found in 13 other strains, nine of which had ANI scores >95% (tblastx >99%), two had ANI scores between 90% and 95% (tblastx 94.5%) and two had ANI scores between 85% and 90% (tblastx 89–94%) (Appendix A). The genome structure and organisation of the 14 prophages was visualised using EasyFig software (Figure 4). A conserved backbone was found of 34–35 kb that comprised genes organised into early (non-structural) and late (structural) regions as found in G4P1. Manual inspection of the lyosgen genomes found a beta-galactosidase gene directly upstream of the G4P1-like prophage insertion site in 11 strains (BCC1170, BCC1186, LA_5_5_30_S1_D2_1, BCC0587, PC082, HI13392, BCC0194, BCC1172, BCC1193, BCC1301 and FL_5_2_10_S1_D0_Repeat). As in G4P1, the beta-galactosidase gene was adjacent to the tail fiber assembly protein, suggesting inclusion in 11 of the G4P1-like prophage genomes.

By using a lower cut-off of 70–75% ANI to compare prophage regions against G4 PHASTER prophage region 10, an additional four G4P1-like variants were identified in strains WP2 (one region), FL_2_3_30_S1_D0 (one region) and BCC0194 (two regions). BCC0194 already had a G4P1-like region, and whether the multiple G4P1-like and G4P1-variant regions resulted from superinfection, prophage duplication, or a combination of both is unknown. When G4P1, G4P1-like and G4P1-variant prophages were aligned, variation in the 34–35 kb shared backbone between prophages could be clearly observed; there was increased diversity in sequence and gene content in the G4P1-variant prophages compared to the G4P1 and G4P1-like prophages (Appendix A; Appendix A).

The transposase proteins identified in G4P1, the 13 G4P1-like and the 4 G4P1-variant prophage genomes clustered together in molecular phylogenetic analysis and were distinct from transposase proteins identified in the Mu-like *Burkholderia* phages BcepMu (NC_005882.1) and ΦE255 (NC_009237.1) (Figure 5). Interestingly, the G4P1 transposase clusters reflected the overall %ANI shared between full length prophage sequences (Figure 5). Transposase proteins sharing the highest sequence similarity to the G4P1 transposases were from phages of *Rhizobium*, *Klebsiella*, *Haemophilus*, *Mannheimia* and another Mu-like *Burkholderia* phage, KS10 (Figure 5; Appendix A). Overall, the protein sets in G4P1, G4P1-like and G4P1-variant prophages were similar, and PHASTER identified that they were more closely related to phages of *Mannheimia, Ralstonia* and *Haemophilus* species than *Burkholderia* phages (Appendix A).

### 3.5. vB_BvM-G4P1 Was Widely Distributed Across the Population Structure of B. vietnamiensis

G4P1 and the G4P1-like prophages were prevalent in *B. vietnamiensis* being found in 17% of the strain collection (14 of 81 strains), and the G4P1-variant found in a further 2 strains. To understand the distribution of the region across *B. vietnamiensis*, phylogenomics was used to determine the population structure of the species. A core-genome phylogeny constructed from 3460 genes revealed that the population split into 6 clades that encompassed 77 out of the 81 strains (Figure 6). Strains carrying the G4P1-like prophage were widely distributed across the phylogenomic tree. Pairs of highly related strains carried the G4P1-like prophage (BCC1193 and BCC1172, BCC1186 and BCC1170, FL_5_2_10_S1_D0_repeat and BCC0194, PC082 and BCC0587, and G4 and BCC1301) but these were found across clades 1, 2, 4 and 5 and were from different isolation sources (cystic fibrosis, environmental, industrial), geographic locations and sequencing projects (Appendix A). The G4P1-variant prophages were found in strains from environmental sources in clades 1, 4 and 6. Strains carrying the G4P1, G4P1-like and G4P1-variant prophages all had evidence of other prophage regions and on average had 3 intact prophages (range: 1–6) and genomes comprising 2.1% prophage material (range: 0.97–3.4%).

## 4. Discussion

As *Burkholderia* are intrinsically antibiotic resistant and almost impossible to eradicate in chronic lung infections in people with CF, there is considerable interest in characterising phages as potential therapeutics [51]. In addition, given the biotechnological interest in *Burkholderia* [8], further genetic tools such as integrative vectors are needed, and understanding prophage biology will enhance the development of these tools [17]. Prophages have been found in the genomes of all *Burkholderia* species that have been investigated except for *B. mallei* [61], and bacteriophages have been isolated from lysogenic strains of *B. ambifaria* [51], *B. cepacia* [62,63], *B. cenocepacia* [51,62,64,65], *B. multivorans* [28,51,62], *B. pseudomallei* [61], *B. pyrrocinia* [51], *B. stabilis* [62], *B. thailandensis* [61,66], *B. vietnamiensis* [28] and members of the closely related *Paraburkholderia* genus [67]. Whilst lytic phages are the optimal choice for phage therapy, genetic modification of temperate phages has been performed to produce non-lysogenic derivatives that have been successfully used therapeutically [68]. The novel phage vB_BvM-G4P1 had broad host range lytic activity against strains of five species of Bcc including *B. dolosa*. Although *B. dolosa* is not the most commonly encountered *Burkholderia* species in CF lung infections, it is one of the most antibiotic-resistant and virulent [69]. In addition, *B. dolosa* strain SLC6, which G4P1 had activity against, was responsible for an outbreak at Boston’s Children’s hospital between 1998 and 2005 that infected over 40 patients and had a high mortality rate [70]. Therefore, further characterisation of vB_BvM-G4P1 is warranted from a therapeutic perspective and as a potential tool to genetically manipulate pathogenic *Burkholderia*.

Apart from vB_BvM-G4P1, the only other phage isolated from *B. vietnamiensis* is NS1 [28], induced from the lysogen *B. vietnamiensis* ATCC 29242 (BCC0587 in this study). In our study, PHASTER determined that BCC0587 harboured a G4P1-like prophage and one other intact prophage, both of which were approximately 35 kb in length (Appendix A). There is no complete genome sequence for NS1 but the phage genome was predicted to be 48 kb in length, 13 kb larger than the regions predicted by PHASTER. Further sequence information for NS1 is, therefore, required to determine the origin of this phage. The novel phage G4P1 was 36.7 kb in length, carried genes for mobility by transposition and had genomic characteristics observed in many Mu-like phages [71]. Mu-like phages have previously been found in the genomes of *Burkholderia*, including DK4 (BcepMu) [64,72] and KS10 [73] which lysogenize *B. cenocepacia* strains, and ΦE255, found in the genome of *B. thailandensis* E255 [61]. As identified by PHASTER, G4P1 and the G4P1-like prophages carry gene sets more closely related to *Mannheimia* (vB_MhM-3927AP2), *Ralstonia* (RS138) and *Haemophilus* (SuMu) phages than *Burkholderia* phages. A recent publication actually identified a similarity between a 13 kb portion of *Ralstonia* phage RS138 and *B. vietnamiensis* G4 chromosome 1 [74] which is likely to be related to G4P1, and the gene content and organisation of G4P1 is highly similar to *Mannheima* and *Haemophilus* Mu-like phages [60]. Whilst the phages of *B. cenocepacia* have been found to encode putative virulence factors, and transduction of antibiotic resistance genes by *Burkholderia* phages has been observed, the overall contributions of phages to virulence and fitness of *Burkholderia* remains largely unknown [28,75,76]. Similarly, the G4P1 prophage does not appear to encode any virulence or antibiotic resistance determinants but has a large number of hypothetical proteins (*n* = 21) of unknown function.

The G4P1-like prophages were common in the genomes of *B. vietnamiensis* being found in 17% of the strain collection. Although phage genomes are renowned for being mosaic in nature [7], recent studies have used genomics to discover the presence of syntenous prophages across different species [77,78], and across different strains of the same species [79,80,81,82]. The present study is the first to report the population genomics of *B. vietnamiensis* and map prophage carriage onto the population. From this analysis we could identify that vB_BvM-G4P1 was not simply localised to a closely related group of strains or single clade, but was widely distributed across the species. As vB_BvM-G4P1 does not appear to encode genes relating to fitness, the mechanism behind its extensive shared carriage is unclear; this occurrence could represent an ancestral integration that has been preserved, or phage transmission between strains in close contact [79,83]. Whether the G4P1-like prophages are inducible and functional remains to be investigated. However, as G4P1 has broad host range lytic activity, there is the possibility that retention of the functional prophage could confer a competitive advantage to the lysogenic population (2). There were also G4P1-variants in two other *B. vietnamiensis* strains and one strain carrying one G4P1-like plus two G4P1-variant prophages. Here, further work is needed to understand the infection processes, immunity systems and evolutionary history of G4P1, G4P1-like and G4P1-variant prophages within *B. vietnamiensis*.

Prophage detection in bacterial genomes remains challenging and the limitations of using only PHASTER for prophage identification are acknowledged [76]. However, PHASTER is an excellent screening tool for examining large numbers of genomes and was applied herein to putatively predict prophage carriage across 81 *B. vietnamiensis* strains, the largest collection of a *Burkholderia* species investigated to date for prophages. All strains had evidence of prophage material, with 59 strains harbouring at least one putative intact prophage. The percentage of genetic material within *B. vietnamiensis* genomes attributed to prophages ranged from 0.1% to 3.4%. This is lower than the percentages found in other bacterial genera (*Escherichia*, *Streptococcus*, *Borrelia*; 10%–20%) [3,83], but close to others (*Salmonella*; 4.9%) [83], and similar to other *Burkholderia* and closely related *Paraburkholderia*. Prophage material was found to comprise up to 3.67% of *B. cenocepacia* genomes (16 strains) [76], up to 2.34% in *B. pseudomallei* (6 strains), 2.00% in *B. thailandensis* (1 strain), 0.55% in *Paraburkholderia xenovorans* (1 strain), up to 4.98% in *B. multivorans* (3 strains) [61] and up to 4% in *Paraburkholderia* genomes (36 strains) [79], although certain *Paraburkholderia* strains can carry up to >10% of their genome as prophage material [67].

## 5. Conclusions

This study exploited *B. vietnamiensis* genomics and phylogenomics for prophage identification, facilitating the discovery of a novel Mu-like phage and its widespread distribution across the species. G4P1 and G4P1-like prophages represent a highly syntenous group, distinct from previously described Mu-like phages found to infect other *Burkholderia* species. G4P1 is inducible from *B. vietnamiensis* G4 and has lytic activity against multiple Bcc species encountered in CF infections, therefore having potential for therapeutic development. Few systematic studies of prophage carriage have been performed for *Burkholderia*, which is likely to change as larger genome collections become available. There is certainly evidence of polylysogeny and considerable untapped potential for the discovery of novel prophages within *Burkholderia* genomes.

## Figures and Tables

**Figure 1 viruses-12-00601-f001:**
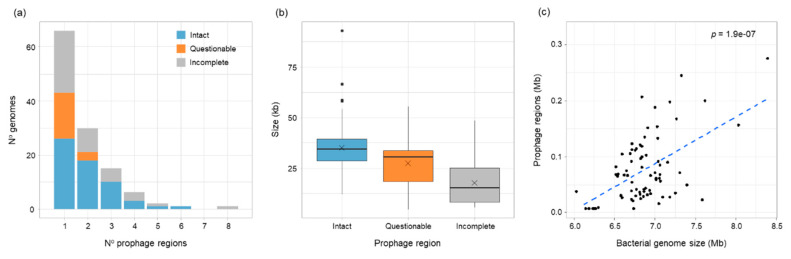
Characteristics of prophage regions identified in 81 *B. vietnamiensis* strains by PHASTER (Phage Search Tool Enhanced Release). (**a**) Distribution of intact (*n* = 115), questionable (*n* = 23) and incomplete (*n* = 81) prophage regions in bacterial genomes. (**b**) Size ranges of prophage regions; boxplots display the mean (x), median, upper and lower quartile, maximum and minimum value, and any outliers. (**c**) Correlation between bacterial genome size and total prophage region content (significant positive correlation; *p* < 0.001).

**Figure 2 viruses-12-00601-f002:**
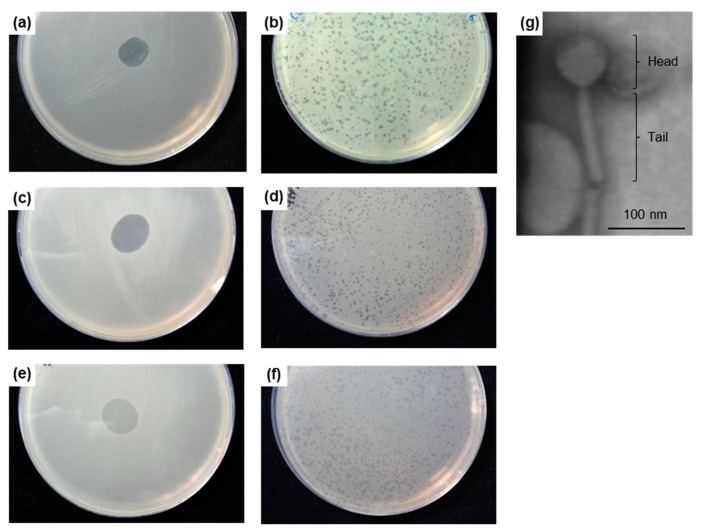
Lytic activity of *Burkholderia vietnamiensis* G4 bacteriophages and morphology of G4P1. G4P1 displayed strong lytic activity against the host strain *B. ambifaria* BCC1212 in a drop test (**a**) and a plaque assay (**b**). G4P2 displayed strong lytic activity against the host strain *B. cenocepacia* BCC1210 in a drop test (**c**) and a plaque assay (**d**). G4P3 displayed weak lytic activity against the host strain *B. cenocepacia* BCC1210 in a drop test (**e**) and a plaque assay (**f)**. Transmission electron microscopy revealed that G4P1 was a tailed phage with an icosahedral head (order *Caudovirales*) approximately 200 nm in length (**g**).

**Figure 3 viruses-12-00601-f003:**
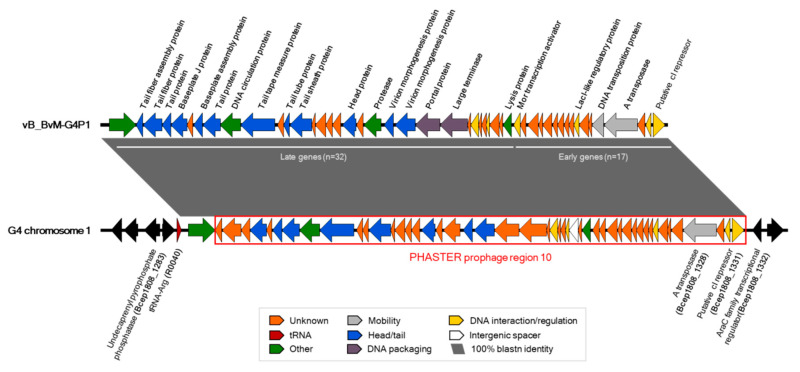
Chromosomal location of the vB_BvM-G4P1 prophage region in *B. vietnamiensis* G4. The G4P1 sequence shared 100% blastn identity with a 36695 bp region on *B. vietnamiensis* G4 chromosome 1. vB_BvM-G4P1 integrates next to a tRNA-Arg (R0040) and spans 49 genes, Bcep1808_1284 to Bcep1808_1331. Two sub-regions are highlighted which contain early (Bcep1808_1315-1331) and late (Bcep1808_1284-1314) genes. This region was identified by PHASTER as prophage region 10 (highlighted by the red box) although inclusion of Bcep1808_1284 (beta-galactosidase) was not predicted by the software. Genes for mobility by transposition are found in the early gene cluster and the “A transposase” protein sequence (highlighted in grey) was used in molecular phylogenetic analysis in Figure 5. Colour-coded functional gene categories are displayed in the key at the bottom and detailed gene annotations are given in Appendix A. Easyfig software was used to visualise the comparison.

**Figure 4 viruses-12-00601-f004:**
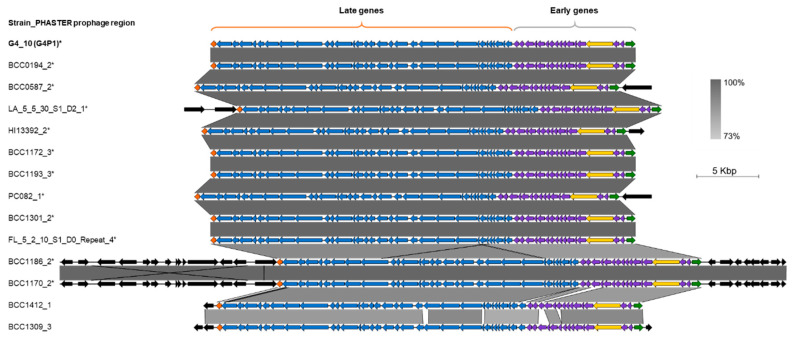
Similarities in the genome organisation of G4P1 and the G4P1-like prophage regions identified by PHASTER. Easyfig was used to visualize the comparison of G4 prophage region 10 (G4P1 without the beta-galactosidase gene) to the 13 G4P1-like prophage regions identified by PHASTER. The comparison revealed a shared backbone of 34–35 kb, starting with a tail fiber assembly protein (orange arrow) and ending with a putative phage cI repressor (green arrow). The region comprised early (grey bracket; purple arrows) and late (orange bracket; blue arrows) genes. Each prophage encoded a transposase gene of 2.2 kb in length within the early region (yellow arrow). Genes outside the shared region are shown by black arrows. Strains have been organized from highest (top) to lowest (bottom) %ANI (average nucleotide identity) similarity to G4 prophage region 10 and prophage region identifiers are shown to the left of the alignment. Genome length is indicated by the scale bar to the right of the alignment. An asterisk (*) next to the strain name indicates the presence of a beta-galactosidase gene upstream of the tail fiber assembly protein in the lysogen genome. The grey vertical blocks between prophage sequences display %blastn shared similarity and the degree of sequence similarity is indicated by the gradient scale.

**Figure 5 viruses-12-00601-f005:**
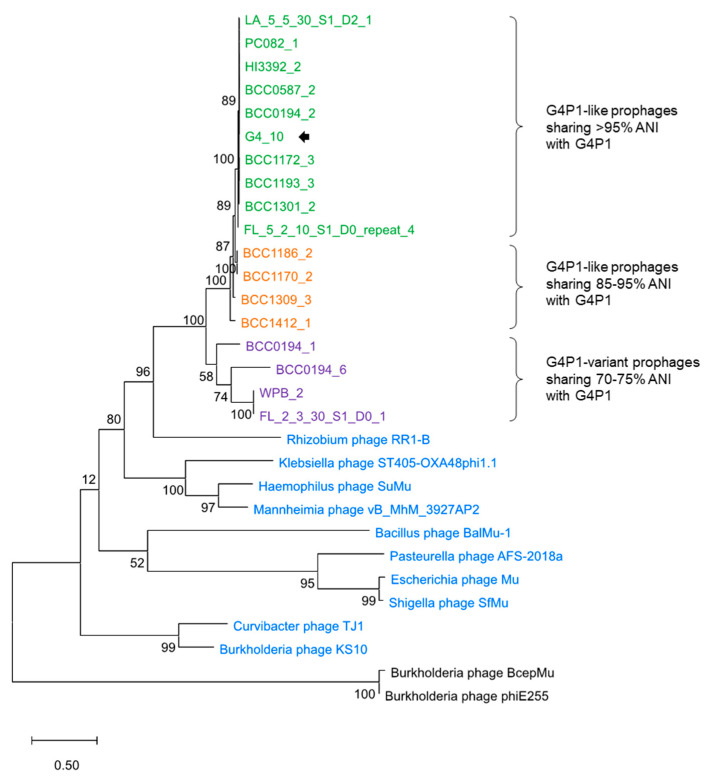
Molecular phylogenetic analysis of prophage transposase proteins. A maximum-likelihood phylogenetic tree infers the evolutionary history of the transposase proteins found in G4P1 (G4_10), the G4P1-like prophages sharing >95% ANI (green) and 85–95% ANI (orange) with G4P1, and G4P1-variant prophages sharing 70–75% ANI G4P1, related transposase sequences in other *Myoviridae* (blue) and Mu-like *Burkholderia* phages (black). *Burkholderia* phage KS10 (blue) is a Mu-like *Burkholderia* phage identified as having a transposase sequence related to the G4P1-like prophages. The position of the *B. vietnamiensis* G4P1 (G4_10) transposase is shown with a black arrow. Bootstrap percentages are shown next to the branches and the scale bar displays the evolutionary distance as number of base substitutions per site.

**Figure 6 viruses-12-00601-f006:**
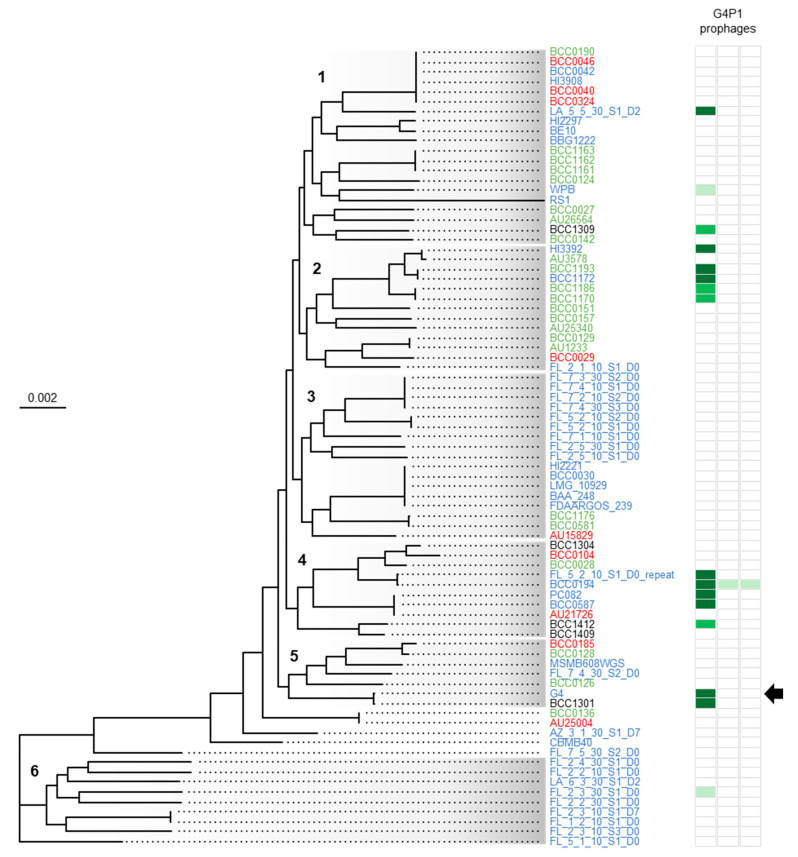
Core-gene phylogeny of 81 *B. vietnamiensis* strains aligned with the presence of G4P1, G4P1-like and G4P1-variant prophages. A Maximum-likelihood phylogenetic tree was generated based on the alignment of 3460 shared ‘core’ genes. The root was determined using a secondary tree containing an outgroup species, *B. ambifaria* AMMD (Appendix A). The scale bar displays the evolutionary distance as number of base substitutions per site. Six clades were identified (grey graded sections; numbers 1–6) that encompassed all but four of the strains (BCC0136, AU25004, AZ_3_1_30_S1_D7 and CBMB40). Strain isolation source is indicated by strain name colour: green is cystic fibrosis, red is clinical, blue is environmental, and black is industrial. The distribution of G4P1 prophages across the species is shown to the right of the core-gene phylogeny. The *B. vietnamiensis* G4 prophage region 10 identified by PHASTER is indicated with the black arrow, dark green blocks represent G4 prophage region 10 and the 9 G4P1-like regions sharing >95% ANI with this region, medium green blocks represent the 4 G4P1-like regions sharing 85–95% ANI and light green blocks show the 4 G4P1-variant regions sharing 70–75% ANI.

**Table 1 viruses-12-00601-t001:** Host range of *B. vietnamiensis* G4 culture supernatant and purified bacteriophage G4P1.

Host Species	BCC#	Alternative Strain Name	Origin (Code)	G4 Culture Supernatant	G4P1
***Burkholderia cepacia complex***					
*B. ambifaria*	BCC0191	HI 2345; J82	Soil (ENV)	+/−	+/−
	BCC0192	Ral-3; R-8863	Rhizosphere (ENV)	+	+
	BCC0197	ATCC 51671	Leaves (ENV)	+	+
	BCC0203	BCF/HG1-A; LMG-P 24640	Environmental (ENV)	+	+
	BCC0338	ATCC 53266; LMG 17828	Roots (ENV)	+	+
	BCC0410	MVP/C1 64	Maize (ENV)	+	+
	BCC1212	MC40-6	Rhizosphere (ENV)	+	+
*B. cenocepacia*	BCC0019	LMG 18829; PC184/NEH4	Cystic fibrosis (CF)	+	+
	BCC1202	AU1054	Cystic fibrosis (CF)	−	−
	BCC1210	MC0-3	Rhizosphere (ENV)	+	−
*B. cepacia*	BCC0001	ATCC 25416; LMG1222-T	Onion (ENV)	−	−
*B. contaminans*	BCC0362	R-9929; CEP0964	Cystic fibrosis (CF)	+/−	+
*B. dolosa*	BCC1343	AU0794	Cystic fibrosis (CF)	Not tested	+
	BCC1356	AU3556; clinical isolate of the SLC6 epidemic strain	Cystic fibrosis (CF)	Not tested	+
	BCC1357	AU1568; clinical isolate of the SLC6 epidemic strain	Cystic fibrosis (CF)	Not tested	+
	BCC1359	AU3960	Cystic fibrosis (CF)	+	+
	BCC1360	AU4298; clinical isolate of the SLC6 epidemic strain	Cystic fibrosis (CF)	Not tested	+
	BCC1361	AU2130; clinical isolate of the SLC6 epidemic strain	Cystic fibrosis (CF)	Not tested	+
*B. lata*	BCC0803	ATCC 17660; LMG 22485T; R-18194; 383	Soil (ENV)	−	−
*B. multivorans*	BCC0005	MA; LMG 18822; C5393	Cystic fibrosis (CF)	−	−
	BCC0011	C1576	Cystic fibrosis (CF)	−	−
	BCC1421	ATCC 17616; LMG17588	Soil (ENV)	−	−
*B. pyrrocinia*	BCC0180	LMG 14191-T	Soil (ENV)	−	−
*B. vietnamiensis*	BCC0027	LMG 18835; PC259; JCM-APRIL93; CEP0040	Cystic fibrosis (CF)	+	+
	BCC0030	LMG 10929; FC0369	Riceroot (ENV)	+	+
	BCC0324	J1742	Non-CF (CLIN)	−	−
	BCC1162	CEP1224	Cystic fibrosis (CF)	−	−
	BCC1304	−	Industry (IND)	−	+/−
	BCC1309	−	Industry (IND)	−	−
**Non-*Burkholderia cepacia* complex**					
*B. gladioli*	BCC0238	MA4	Cystic fibrosis (CF)	−	−
*B. thailandensis*	BCC0779	LMG 20219; ATCC 700388; E264	Soil (ENV)	−	−
***Paraburkholderia***					
*P. phymatum*	BCC1607	LMG 22487; PsJN	Environmental (ENV)	−	−
*P. phytofirmans*	BCC1604	LMG 21445	Environmental (ENV)	−	−
*P. graminis*	BCC0774	ATCC 700544; LMG 18924	Soil (ENV)	−	−

Footnotes: (+) with green background, strong lytic activity; (+/−) with blue background, weak lytic activity; (−) no activity; ENV, environmental; CF, cystic fibrosis; IND, industry.

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
