# Peer review of "A Novel Inducible Prophage from Burkholderia vietnamiensis G4 Is Widely Distributed across the Species and Has Lytic Activity against Pathogenic Burkholderia"

_viruses, 2020, doi:10.3390/v12060601_

Round 1

Reviewer 1 Report

Manuscript by Weiser et al. describes prophages in Burkholderia vietnamiensis which is part of the Burkholderia cepacia complex (Bcc) that are linked with lung infections in CF patients. In their work the authors use bioinformatic and experimental approaches. The work is valuable addition to the Burkholderia prophage field and also increases the possibilities for identification of prophages from other bacterial species. Here, prophages were searched using PHASTER (and authors acknowledge its limitations) from 81 B. vietnamiensis genomes (of which 35 were sequenced in this work). In addition, one functional prophage is characterized giving solid base for the bioinformatic results. 

More specific comments

This is maybe a matter of taste, but the induced prophage is named in the abstract and introduction (vB_BvM etc..) – Maybe leave the name out or call it as G4P1 (which is used in parallel) and introduce the long name in the Results as it is in line 285.

Line 96: In brief, describe at least the sequencing platform

Lines 145-161: What strain was used as an indicator strain for prophages? What strain was used to produce the phage lysates? (please add them here already)

Line 166: Was the positive result confirmed also for individual plaques using dilution series? If not, how can you be sure that the clear zone is phage origin and not other inhibiting factor?

Figure 2 g: Do you have other TEM images that have the phage. The quality of this one is not very good, maybe another angle could help in addition.

Line 421: How about the genome sequence of BCC0587? Is it possible to predict NS1 from that?

Reviewer 2 Report

In this manuscript, Weiser et al. apply PHASTER to 81 sequenced B. vietnamiensis strains to identify multiple putative lysogenic phages. They identify a novel lysogenic phage that is broadly distributed among Burkholderia strains.

Overall the paper is fine, but could be made clearer.

Minor issues:
Lines 234-235: This statement is misleading, as while being able to plaque a prophage does require it to be spontaneously inducible, it is not the case that all spontaneously inducible phages will plaque. The sentence should be revised. To that point, what about the other prophage regions that they were not able to plaque? Did the authors do PCR to assess whether those regions of DNA were enriched in the supernatant?

Lines 248-250: The authors describe the induction frequencies of each of these phages, but they do not describe how they can specifically plaque each of these phages in a comparable manner. While they say GP2 has a different host tropism than GP1, plaquing different phages on different host strains is not an accurate way to determine comparable phage titers, as the host strains might contain different anti-phage systems that could restrict plaquing. The best way would be to compare the release by qPCR of the supernatant, along with data about plaquing on the same, or at least very similar strains.

Figure 4/Figure S3: It is not clear what this figure is displaying, (I'm guessing the purple/green lines are the conservation?). To display the genome organization, the figure should be reformatted like other phage discovery/evolution papers do. (Please see Figures 1 and 4 of Yutin et al. Nature Microbology 2017 and Figure 2 of Pope et al. Plos One. 2011 for how to display the conservation of genome organization between multiple genomes)

Table 1: The host range of this phage is interesting, but this data is presented without context. Specifically, the authors test multiple isolates within the BCC complex, but as a non-Burkholderia expert, I cannot assess how simmilar these strains are to each other. The authors imply that the strains used to test the host range of the phage, so it would be very useful to not only compare how related these strains are and describe whether phylogenetic relatedness determines host tropism, but also to use comparative genomics to identify bacterial genes that enable infection.
